# Inuit knowledge of Arctic Terns (*Sterna paradisaea*) and perspectives on declining abundance in southeastern Hudson Bay, Canada

Dominique A. Henri[1]*, Laura M. Martinez-Levasseur[1], Salamiva Weetaltuk[2], Mark L. Mallory[3], H. Grant Gilchrist[4], Frankie Jean-Gagnon[5]

1 Wildlife Research Division, Environment and Climate Change Canada, Montréal, Québec, Canada, 2 Local Nunavimmi Umajulivijiit Katujaqatigininga, Kuujjuaraapik, Québec, Canada, 3 Biology Department, Acadia University, Wolfville, Nova Scotia, Canada, 4 Wildlife Research Division, National Wildlife Research Centre, Environment and Climate Change Canada, Ottawa, Ontario, Canada, 5 Nunavik Marine Region Wildlife Board, Inukjuak, Québec, Canada

* dominique.henri@canada.ca

**Data Availability Statement:** The data used in this study are restricted by the Local Nunavimmi Umajulivijiit Katujaqatigininga of Kuujjuaraapik due

## Abstract

The Arctic Tern (*Sterna paradisaea*; *takatakiaq* in Inuttitut) breeds in the circumpolar Arctic and undertakes the longest known annual migration. In recent decades, Arctic Tern populations have been declining in some parts of their range, and this has been a cause of concern for both wildlife managers and Indigenous harvesters. However, limited scientific information is available on Arctic Tern abundance and distribution, especially within its breeding range in remote areas of the circumpolar Arctic. Knowledge held by Inuit harvesters engaged in Arctic Tern egg picking can shed light on the ecology, regional abundance and distribution of this marine bird. We conducted individual interviews and a workshop involving 12 Inuit harvesters and elders from Kuujjuaraapik, Nunavik (northern Québec), Canada, to gather their knowledge of Arctic Tern cultural importance, ecology, and stewardship. Interview contributors reported a regional decline in Arctic Tern numbers which appeared in the early 2000s on nesting islands near Kuujjuaraapik. Six possible factors were identified: (1) local harvest through egg picking; (2) nest disturbance and predation; (3) abandonment of tern nesting areas (i.e., islands that have become connected to the mainland due to isostatic rebound); (4) climate change; (5) natural abundance cycles within the Arctic Tern population; and (6) decline of the capelin (*Mallotus villosus*) in the region. Recommendations from Inuit contributors related to Arctic Tern stewardship and protection included: (1) conduct more research; (2) let nature take its course; (3) conduct an awareness campaign; (4) implement an egg picking ban; (5) coordinate local egg harvest; (6) start 'tern farming'; (7) protect Arctic Terns across their migration route; and (8) harvest foxes predating on terns. Our study highlighted complementarities between Inuit knowledge and ecological science, and showed that Inuit harvesters can make substantial contributions to ongoing and future Arctic tern research and management initiatives.

to potentially sensitive information regarding Inuit land-use and harvesting activities. Please address data inquiries to: Ms. Salamiva Weetaltuk, Manager, Local Nunavimmi Umajulivijiit Katujaqatigininga of Kuujjuaraapik; Phone: (819) 929-3722/3111; Fax: (819) 929-3723; Email: salamiva@gmail.com.

**Funding:** This study was funded by the Nunavik Marine Region Wildlife Board (https://nmrwb.ca/) and Environment and Climate Change Canada (http://ec.gc.ca/default.asp?lang=En&n=BD3CE). The funders had no role in study design, data collection and analysis, decision to publish, or preparation of the manuscript.

**Competing interests:** The authors have declared that no competing interests exist.

## Introduction

The Arctic Tern (*Sterna paradisaea*; *takatakiaq* in Inuttitut) breeds in the circumpolar Arctic, and winters in marine waters near Antarctica [1–3]. This trans-equatorial migrant undertakes the longest known annual migration of any species on Earth [1, 2]. Arctic Terns are plunge-diving and surface-dipping seabirds, which feed on the top half meter of the water column, mainly on small fish, such as capelin (*Mallotus villosus*), sandlance (*Ammodytes* spp.) and Atlantic herring (*Clupea harengus*), on crustaceans, such as euphausiid shrimp (*Meganyctiphanes norvegica*), and on large zooplankton [4–7]. The breeding range of the Arctic Tern extends from the northern coast of Greenland (84˚N) to the eastern coast of North America (42˚N) [3]. Terns breed scattered in pairs on coastal tundra and in colonies on offshore islands typically inaccessible to mammalian predators [8, 9]. If conditions are adequate (i.e., absence of snow and predators), they lay one or two eggs, occasionally three, in ground nests, with no significant differences in egg numbers across latitudes within the American Arctic [4, 7, 10]. Arctic Terns generally display high fidelity to breeding colonies [11], although they can move between breeding sites in response to disturbance such as predation events or food scarcity [5, 7].

The Arctic Tern is currently listed as a species of "Least Concern" by the International Union for Conservation of Nature (IUCN) Red List, mainly because of its large range and population size, estimated at over two million individuals worldwide [12]. Yet, the overall Arctic Tern population trend is apparently decreasing [12]. Although reasons behind this decline might differ across their geographical range, predation by Arctic foxes (*Vulpes lagopus*), polar bears (*Ursus maritimus*), multiple bird species and humans, as well as changes in prey abundance have led Arctic Terns to abandon some breeding colonies [7, 12, 13]. To date, limited scientific information exists on Arctic Tern ecology and abundance in the Canadian Arctic [8, 9, 14]. Establishing tern population trends through colony surveys is difficult in the Arctic due to large variation in colony attendance between breeding seasons [15] and given the high cost and complex logistics associated with field research in remote locations [16, 17]. Although Arctic Terns breed in Nunavik (northern Québec, Canada) during summer months, limited scientific information exists on their ecology and abundance in this region [9]. Recently, Inuit from various communities in Nunavik have reported that Arctic Terns are in decline and have expressed the need for more research on this species. The Arctic Tern is a culturally important species to Nunavik Inuit who harvest tern eggs for subsistence.

Inuit Traditional and Local Ecological Knowledge (TEK/LEK) could contribute information on the regional ecology, population trends, and distribution of Arctic Terns along the coast of Québec, Canada. TEK/LEK held by Indigenous peoples can be conceptualized as an 'evolving practice-knowledge-belief system transmitted over generations and focused on ecological relationships' [18–20]. The term Local Ecological Knowledge (LEK), which designates placed-based ecological knowledge acquired over a lifetime, has also been used [21–25]. Collectively, Indigenous knowledge systems can contribute qualitative information on multiple aspects of wildlife ecology, particularly for species that are wide ranging in remote environments and harvested by local communities [19, 26–28].

In this context, our study objectives were to: (1) gather and document Inuit knowledge of Arctic Tern cultural importance and ecology, as well as Inuit perspectives related to Arctic Tern stewardship and management; (2) compare documented Inuit knowledge with available scientific information about Arctic Terns; and (3) identify strategies and future opportunities for community-based monitoring and research on Arctic Terns in Nunavik. To do so, we conducted interviews and a workshop with 12 Inuit harvesters and elders from Kuujjuaraapik, Nunavik (northern Québec), Canada, who were all actively engaged in Arctic Tern egg picking.

## Materials and methods

Institutional review boards from the Wildlife Research Division of Environment and Climate Change Canada and the Nunavik Marine Region Wildlife Board have approved this study. Written consent was obtained from all project contributors. The study followed ethical guidelines from the Canadian Institutes of Health Research, the Natural Sciences and Engineering Research Council of Canada and the Social Sciences and Humanities Research Council of Canada for conducting research involving Indigenous peoples in Canada [29].

### Inuit knowledge

Throughout this paper, we employ the expression 'Inuit knowledge' to refer to the knowledge acquired by Inuit contributors through their lifetime and intergenerational transmission [20, 28]. Other terms exist such as 'Inuit Qaujimajatuqangit' (IQ)–literally, "that which has been long known by Inuit" [30], which commonly refers to traditional Inuit knowledge, culture and societal values [31–34]. Traditional and Local Ecological Knowledge can also be conceptualized as one aspect of Inuit knowledge focused on ecological relationships [24, 25]. Here we preferred employing 'Inuit knowledge' given that this expression is broader in scope than IQ and TEK/LEK and has been frequently employed in recent ecological research in Nunavik [35, 36].

### Study area

This study was undertaken jointly by government and academic researchers in partnership with the Nunavik Marine Region Wildlife Board, the Local Nunavimmi Umajulivijiit Katuja-qatigininga (LNUK; or local Hunters, Fishers, and Trappers Association [HFTA]) of Kuujjuar-aapik, and SIKU/Arctic Eider Society (S1 File). The LNUK as well as the Regional Nunavimmi Umajulivijiit Katujaqatigininga (RNUK; or regional HFTA) gave permission, and contributed advice and support to carry out this work. Field research was conducted in the community of Kuujjuaraapik, Nunavik (northern Québec), Canada (55˚28'N, 77˚76'W; Fig 1). Out of its population of 686 inhabitants, 505 reported themselves to be Inuit and 110 to be First Nations (Cree-Montagnais) [37]. Inuit started to settle in the village in the late 1930s [38].

In July 2018, following support from the RNUK and LNUK, we initiated this study with the community of Kuujjuaraapik, where there was strong local interest, concern for the species, and knowledge of Arctic Tern ecology. Later that same year, we conducted semi-directive interviews with both individuals and groups among Kuujjuaraapik residents identified to us by the LNUK as 'local experts'; that is, "persons recognized by their peers as knowledgeable" about Arctic Tern ecology [39, 40].

### Interviews

We conducted 1–2 h interviews with 11 local Inuit experts, including four women and seven men ranging between 18 and 70 years old. Experts were interviewed individually or in groups of two, according to their preference. All were engaged in Arctic Tern egg collecting. Main themes that were discussed included: (1) the importance of Arctic Terns for Nunavik Inuit; (2) aspects of Arctic Tern ecology (e.g., migration, reproduction, feeding, predation, habitat, distribution, abundance trends); and (3) Arctic Tern stewardship and management by local Inuit residents (S2 File). In all cases, interviewees received an invitation letter, signed a consent form describing their rights and conditions for release of recorded information (S3 File), and received an honorarium for their time. Throughout this paper, the words "interviewee" and "contributor" are used synonymously and interchangeably.

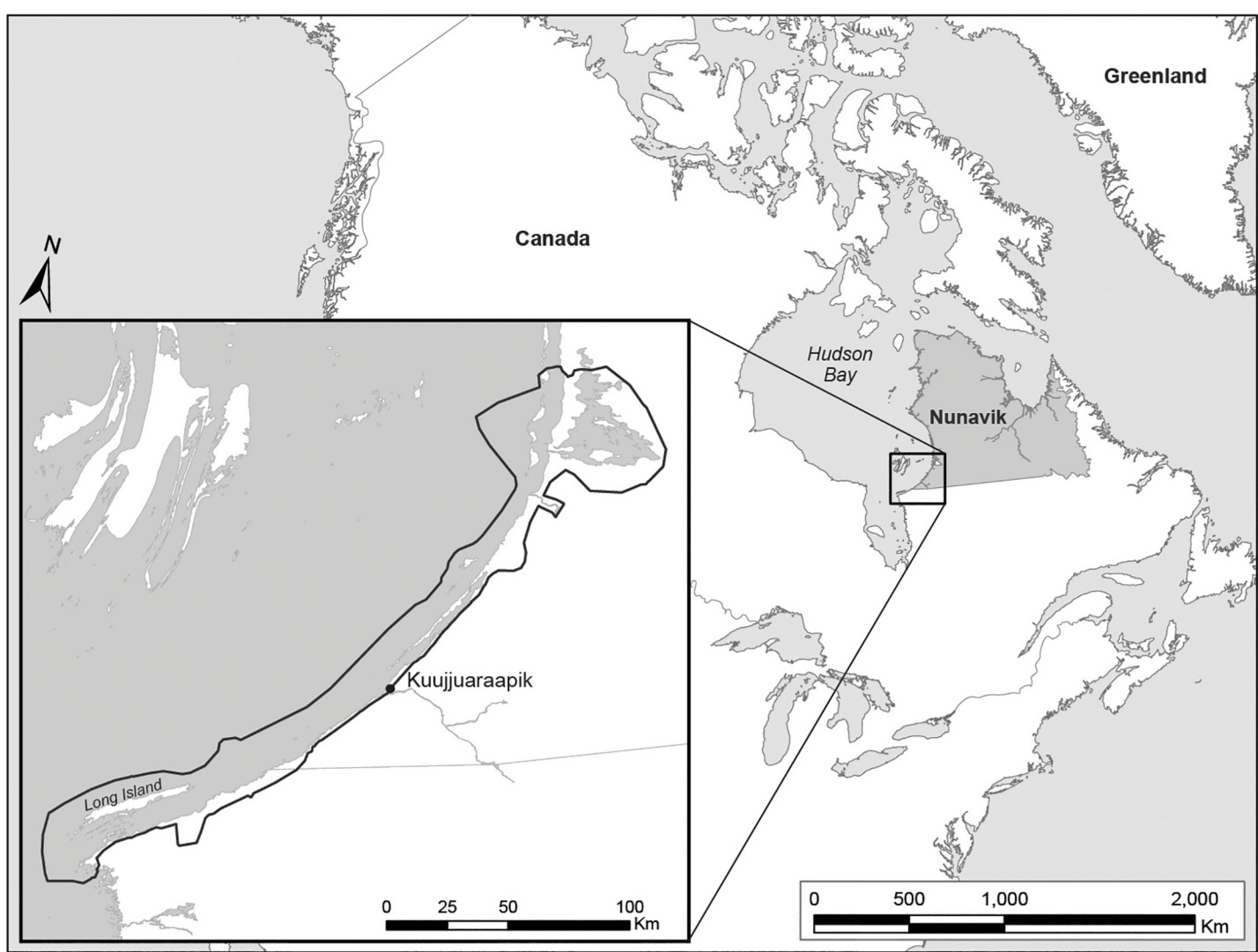

**Fig 1. Study area and common area of Arctic Tern observation among interview contributors from Kuujjuaraapik (n = 11).** The area located within the thick grey line indicates contributors' common area of observation, i.e. area where interviewees conducted land-based activities (boating, camping, fishing, hunting, and egg picking) in summer and early fall, when terns were present in the region, and for which they had direct observational knowledge and experience.

Interviews were audio-recorded with permission and conducted in contributors' language of choice (Inuttitut or English) with assistance from one local interpreter. Spatial information was recorded on 12 topographic maps created by the Avataq Cultural Institute (scale 1:35,000). The information was then transcribed digitally using the geographic information system, Arc-Map 10.4.1. (digital vector datasets: NRCan-National Topographic Database). Contributors drew points (e.g., camp sites, hunting locations), lines (e.g., boat routes) or polygons (e.g., Arctic Tern colonies) onto plastic transparencies placed over the maps [25, 41]. They also drew the geographic areas where they conducted land-based activities in summer and early fall, which corresponded to the period when terns are present in the region, and for which interviewees had direct observational knowledge (Fig 1). We refer to these geographic areas as 'common area of observation' [25]. Establishing a common area of observation among interviewees allowed us to distinguish between (1) areas which had been visited by Inuit harvesters within which tern presence or absence could be assessed, and (2) areas that had not been

visited and for which harvesters could not provide direct observations [25]. Following the interviews, all audio-recordings were transcribed, entered and coded using NVivo Pro 11 (Version 11, 2012), and subsequently analyzed using thematic content analyses [42].

### Results validation

Following interviews and preliminary analyses, we returned to the community and held a validation workshop. Fifty-five percent of interview contributors (6 out of 11) and one additional Kuujjuaraapimiut (Kuujjuaraapik resident)–identified by the LNUK as a 'local expert' on Arctic Terns–attended the two-day validation workshop [24, 42]. Employing a validation workshop allowed contributors to provide feedback on preliminary results, confirming information that was unclear or contributing new or missing information. It also allowed contributors to achieve further consensus about certain topics such as regional abundance of Arctic Terns over time [24, 34]. Feedback gathered during the validation workshop was incorporated into our final analysis. Final results were presented to the community in 2020 (S4 File).

### Limitations

Some limitations and potential biases applicable to this study should be acknowledged. First, the information collected through interviews represented a partial sample of all knowledge held about Arctic Tern ecology in Kuujjuaraapik. Responses received during interviews might have been influenced by the level of familiarity of the interviewer with the local culture, the personality and gender of the interviewer, the lack of recall of specific factual information by interviewees, and the loss of information through the translation process [43]. In addition, given that knowledge holders were also resource users, respondents might have been reluctant to reveal proprietary or sensitive knowledge. However, we felt interviewees were comfortable sharing their knowledge because of the presence of strong local support for this project, and the shared concern between Inuit, scientists, and wildlife managers regarding a decline in Arctic Tern abundance observed in the region.

## Results

Our results are divided into three main sections: (1) importance of Arctic Terns to Inuit; (2) Arctic Tern ecology; and (3) Arctic Tern stewardship and management. When presenting results, we separated factual observations made by interviewees from inferences, and direct observations made by contributors from those they reported from other hunters, elders, or relatives [28, 44]. Similarly, we distinguished contemporary knowledge from knowledge passed down from earlier generations. We also provided representative quotes from contributors to illustrate themes discussed during interviews and the subsequent validation workshop. The anonymity of all interviewees was preserved.

### Importance of Arctic Terns to Inuit

**Nomenclature.**    In Kuujjuaraapik, Arctic Tern is called *takatakiaq* in Inuttitut (plural: *takatakiait*). This word mimics their calls. Some contributors also reported that in Nunavik communities situated north of Kuujjuaraapik and in Nunavut, Arctic Terns may also be called *immiqutailaq*.

**Egg picking.**    All interviewees reported that collecting the eggs of Arctic Terns is important to Nunavik Inuit for culture, community well-being, and sustenance. Kuujjuaraapimiut enjoy collecting and sharing Arctic Tern eggs with relatives and friends. Arctic Tern eggs are considered a delicacy by Nunavik Inuit; eggs are mostly eaten boiled, but can also be fried or sometimes eaten raw.

Freshly laid eggs are the best. We don't like to eat the ones with the embryo. (Interviewee #7)

After we pick eggs, we cook the eggs there [on the islands]. Because, especially if the water's rough, they're going to all crack on the boat. If we bring [. . .] them home, we cook and share them with five here, five there, because we know everybody's craving for them. (Interviewee #8)

All contributors agreed that egg picking takes place during a two-week period in early July. Inuit harvesters generally collect eggs opportunistically while on hunting or fishing trips.

It's a tradition whenever you go out hunting at the beginning of July. We'd go to an island and have a short lunch and boil some eggs. We don't take the whole island's eggs, we just have what we needed for that meal. (Interviewee #7)

In total, 55% of contributors (6 out of 11) highlighted that, in addition to be a source of food, egg picking contributes to community well-being by allowing people to be on the land, and to work together.

We take the elders out on an egg picking day [. . .] Every ice break, [the local Landholding Corporation] takes elders out on egg picking to relieve their craving, and re-live what they used to do since they were children themselves. (Interviewee #8)

Little kids when they look for eggs it's very soothing for them. [They] don't think of anything else but to look for eggs. (Interviewee #3)

One interviewee spoke about the importance of showing respect to Arctic Terns, a value that has been passed down through generations.

Well, I guess it was something that was passed down to us. 'Don't hurt the little terns because they're going to provide you some eggs' kind of mentality. So we always respected that, we never shot them for target practice, we never killed them for no reason [. . .] We were told to respect them. (Interviewee #1)

Finally, 73% of contributors (8 out of 11) were concerned that knowledge about Arctic Terns was not passed on to the younger generations.

Today's generation I call them the 'iPod generation'. They're not getting enough traditional knowledge, [. . .] not just [about] Arctic Terns, but camping and being out on the land altogether. What they do know probably comes from Google, but not from firsthand knowledge from the elders passing it down to them because [young people] are not out there anymore. (Interviewee #1)

The last few years I have noticed that we've lost elders that used to teach us, younger generations, what to hunt, how to hunt, where to go. (Interviewee #10)

Some of the young ones would have [less] knowledge today because there's not much of a hunting [culture] anymore for some young people who have no equipment [. . .] Before, when we were young, everybody used to have their own canoes, their own kayak, their own equipment to go out. But today it's completely different. (Interviewee #6)

However, perspectives on intergenerational knowledge transmission varied, with 18% of contributors (2 out of 11) saying that young people who have the opportunity to go out hunting with their family still learn about Arctic Terns from their elders.

**Arctic Terns as indicators.** Kuujjuaraapik residents use Arctic Terns as an indicator of environmental conditions and wildlife presence. One interviewee stated that the presence of terns in 'early summer' (i.e., June) indicates that warmer weather is coming. Eighteen percent of interviewees (2 out of 11) mentioned that Arctic Terns diving at sea can act as indicators of good seal hunting and fishing areas. These contributors explained that Arctic Terns and seals both feed on small fish, and, as such, can often be found feeding in the same areas.

Arctic Terns help us when we're seal hunting [. . .] When we see Arctic Terns going like this [flying up and down] that means a seal is feeding around. (Interviewee #6)

We sort of use them as radar [to find seals]. (Interviewee #1)

You can tell there is lot of fish in that lake when there's the Arctic Tern that lay their eggs there. It's an indicator that there's fish there. (Interviewee #7)

The observation of Arctic Terns diving in terrestrial areas, particularly over their nesting areas, can also warn of the presence of predators such as foxes, wolves or polar bears.

When we are going to the island and if the terns are going like this [flying up and down] on the land that means it's either polar bear or wolf. (Interviewee #6)

Overall, contributors commented about the value of Arctic Terns for the ecosystem.

I think they do play an important role in our ecosystem. Any little species that lives on the Earth, I guess, is important [. . .] And recently I've heard that [species] are disappearing at a very alarming rate and I'm afraid that the tern might be one of them. (Interviewee #1)

## Arctic Tern ecology

**Migration and reproduction.** In total, 36% of contributors (4 out of 11) explained that Arctic Tern arrive around Kuujjuaraapik in late June and early July when the weather is warm enough, and the ice and snow are melting (Fig 2). They arrive in the region progressively in small groups at a time of year when Kuujjuaraapik residents are deploying their fishing nets, cleaning up their camps, and picking eggs from other bird species. According to 27% of contributors (3 out of 11), terns may choose to nest on different islands from year to year–preferring those with no terrestrial predators such as foxes–but return to the same general area.

If there's an animal [a predator] on that island, [terns] will not lay their eggs [there] for that summer. But next summer if there is no animal there, they would lay their eggs. They don't abandon their laying egg campsite forever. (Interviewee #7)

One interviewee reported hearing elders say that terns go back to nest on the island where they were hatched.

Two contributors (18%; 2 out of 11) highlighted that when Arctic Terns arrive in early summer, they wait until it is warm enough to lay their eggs. Terns prefer laying their eggs when the ice and snow have melted, which keeps away terrestrial predators from nesting islands, and gives terns access to open water for feeding.

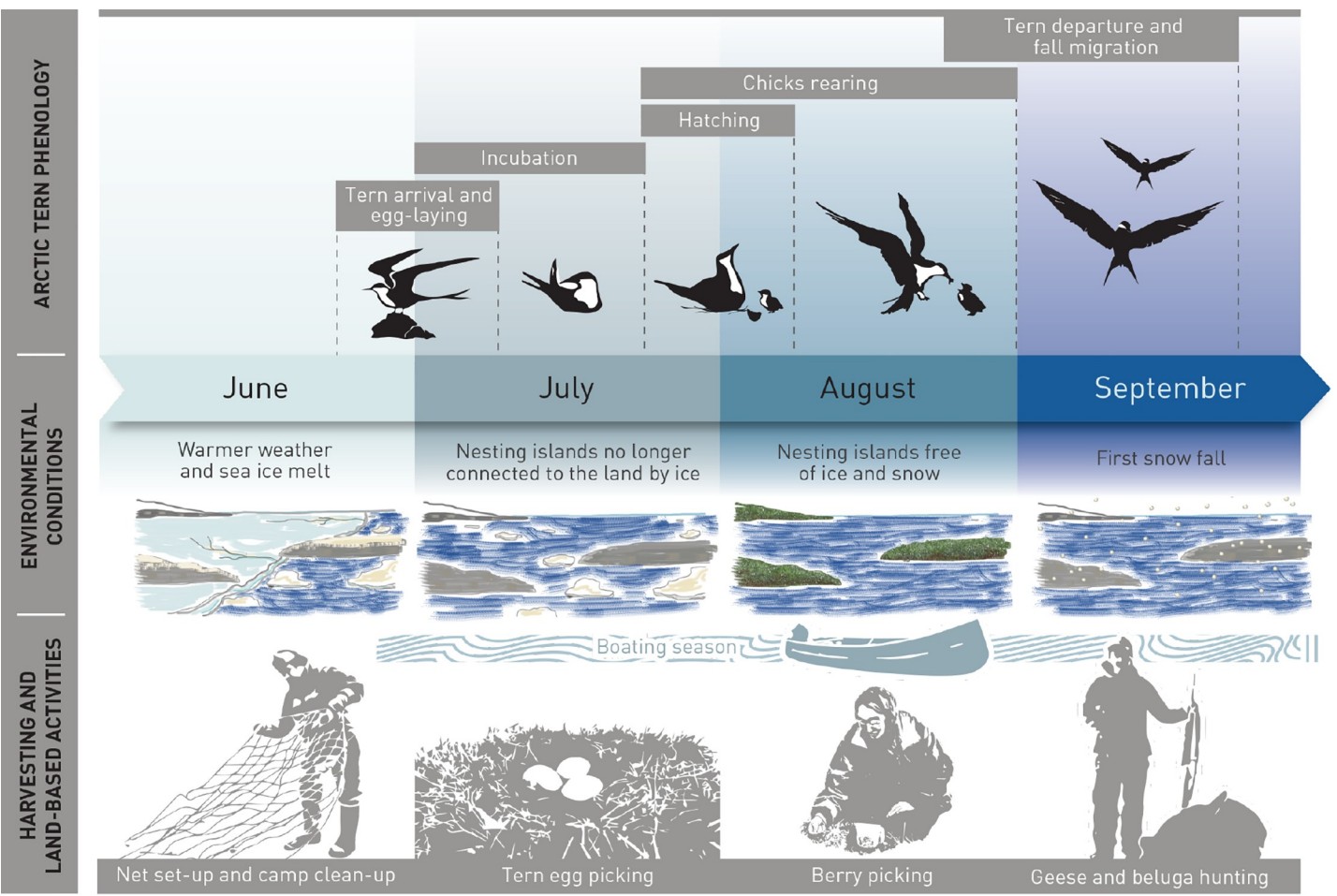

**Fig 2. Arctic Tern breeding and migration phenology calendar.** Information on Arctic Tern phenology provided by Kuujjuaraapik contributors (n = 11) is presented above the timeline. The two rows under the timeline highlight associated environmental conditions and land-based activities relevant to tern observation by Inuit harvesters around Kuujjuaraapik. Timing of Arctic Tern life cycle events, environmental conditions and land-based activities can show interannual variations.

When you see the terns [in early summer] you know it's going to start getting warmer. They start gradually appearing, and they hang around for a while before they actually start having eggs. I've noticed that they're reluctant to lay their eggs when they see patches of ice anywhere on the [land] or on the sea. (Interviewee #1)

Around Kuujjuaraapik, Arctic Terns start laying their eggs in late June (Fig 2). They also lay eggs in early July. Two contributors (18%; 2 out of 11) explained that Arctic Tern typically lay two to three eggs in one summer.

They normally have two or three eggs during the summer [. . .] Just by looking at them [in early fall] and their small down feathers that they have, they tell you exactly how young or how old they are. (Interviewee #1)

One interviewee had observed up to four or five eggs in one nest but mentioned this was rare. Another contributor stated that terns tend to lay more eggs on rainy days. Two interviewees (18%; 2 out of 11) also highlighted that Arctic Terns are able to lay new eggs to replace the ones that were picked or depredated.

All interviewees highlighted that Arctic Terns prefer to lay their eggs on small islands located in coastal saltwater areas. Two (18%; 2 out of 11) also commented that terns can also nest on islands located in freshwater lakes (inland). Contributors however acknowledged that their observations of terns nesting on the mainland was limited as they mainly travel through coastal areas by boat over summer months during the open water season (Fig 2).

Interviewees explained that terns prefer nesting on flat areas with small pebbles or gravel, and in places that have a bit of grass: "They lay their eggs on gravel because [. . .] the gravel is heated by the sun and they need warm weather" (Interviewee #7). Arctic Terns can also lay their eggs on sand, short grass, moss, small driftwood, and on big flat rocks.

Contributors had observed terns co-habiting with other bird species on nesting islands, such as eiders, gulls, Black Guillemot (*Cepphus grylle*), Long-tailed Duck (*Clangula hyemalis*), and Canada Goose (*Branta canadensis*). Tern interactions with these birds were reported to be 'generally peaceful', although gulls can depredate tern eggs.

In the Kuujjuaraapik area, one interviewee observed that chicks generally start appearing around the first week of August which is three weeks after eggs were first laid (Fig 2). Twenty seven percent of contributors (3 out of 11) also reported that fall migration of terns out of the region takes place the first two weeks of September when the weather starts to get colder and before the first snowfall (Fig 2). At that time of year, Kuujjuaraapik residents harvest beluga (*Delphinapterus leucas*) and geese and report observing terns leaving in small groups, similarly to the spring migration.

**Prey and predators.**   Interviewees discussed interactions between Arctic Tern, its prey and its predators (Fig 3). Arctic Terns mostly feed in saltwater areas. They feed mainly on fish, such as capelin (*quuliliraq*, *qulliliraq* or *quliigak*) and *ammayait* (English translation unknown). Two interviewees (18%; 2 out of 11) observed that terns also feed on sandlance, a long thin fish which seems new in the region.

> There's like a fish species here [sandlance] [. . .] It's like a string, very thin. I've never seen that anywhere in the books or anywhere. I think it might be a new species or something [. . .] Whales have that in their stomachs [. . .] and seals eat that too. (Interviewee #7)

One contributor said that terns can feed on freshwater 'shrimps' (*kinguit*; gammarids). Another reported that they can feed on mussels "when there is nothing else [to eat]" (Interviewee #8).

Eighteen percent of contributors (2 out of 11) highlighted that terns look for seals to find places where there are fish to eat.

> Normally you would see some seals, terns and seagulls, all feeding at the same time [. . .] I think they work together [to feed] somehow. (Interviewee #1)

> When the seals are feeding, the Arctic Terns often start feeding [on] what the seals are eating [. . .] [Terns] coordinate their egg laying with the seal feeding. (Interviewee #7)

According to interviewees, Arctic Tern eggs are eaten by humans, Arctic foxes, polar bears, wolves (*Canis lupus*), Glaucous Gulls (*Larus hyperboreus*), Herring Gulls (*Larus argentatus*), Jaegers (*Stercorarius* spp.) and Common Ravens (*Corvus corax*). According to 36% of interviewees (4 out of 11), the Common Raven is a new predator for terns around Kuujjuaraapik. Ravens did not use to feed on tern eggs in the past; their behaviour has changed in recent years.

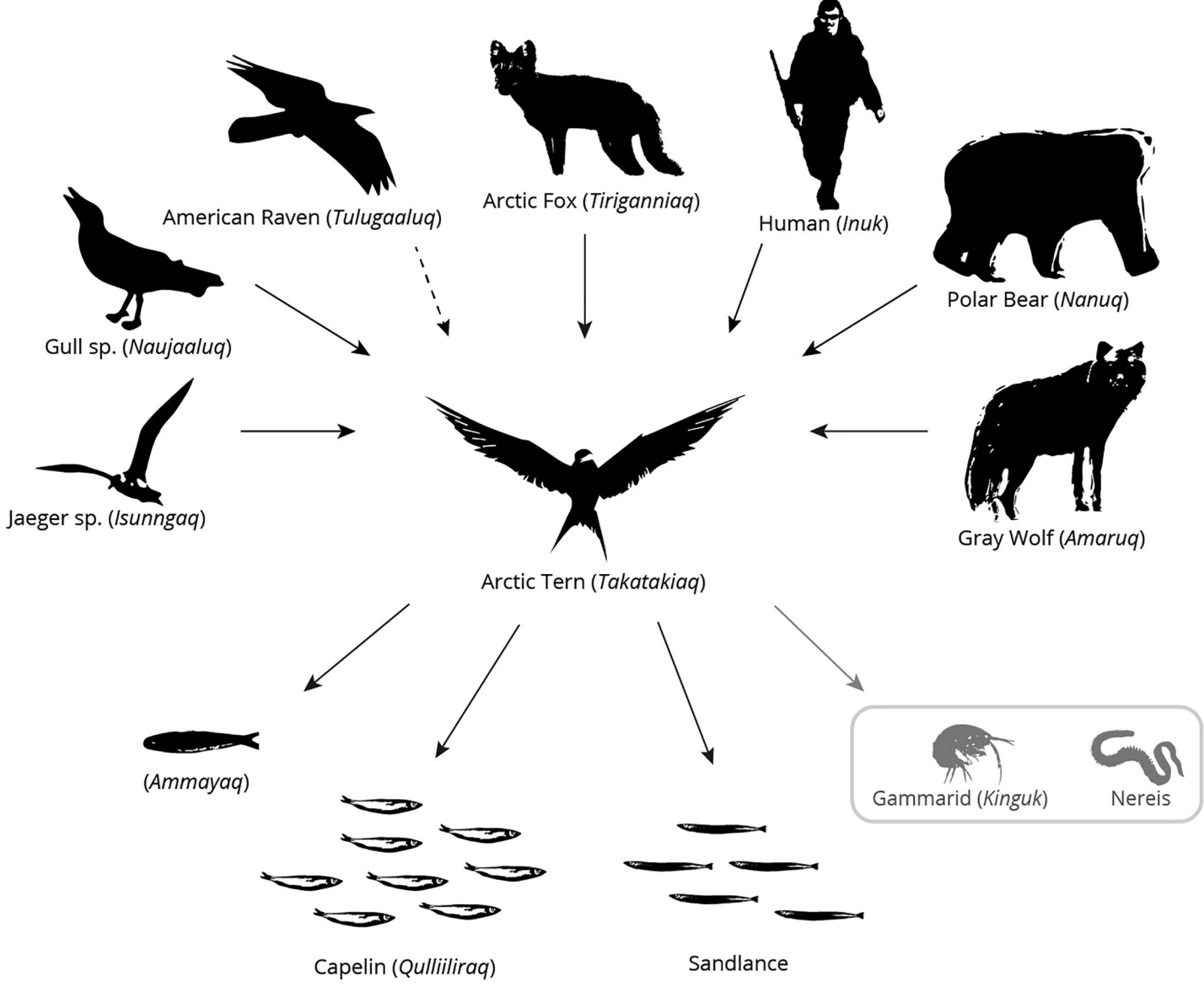

**Fig 3. Arctic Tern predators (upper part) and prey (lower part) identified by project contributors (n = 12).** Prey and predators are presented in both English and Inuttitut (within parentheses) when possible. The dotted arrow identifies a new predator (i.e., the raven). Species in the grey rectangle are those identified by contributors as potential prey for young Arctic Terns, although these prey-predator relationships were not directly observed by contributors.

In the past, there were a few [ravens] [. . .] They somehow invade the islands now. Before [ravens] were not able to fly over water but they learned that. (Interviewee #6)

Two contributors (18%; 2 out of 11) explained that terns, chicks especially, can also be eaten by birds of prey, such as Peregrine Falcons (*Falco peregrinus*) and Merlin (*Falco columbarius*).

One interviewee explained that a decline in Arctic Tern abundance observed around Kuujjuaraapik has been limiting Arctic Terns' ability to protect themselves against avian predators, and that gulls, eiders, and ravens had 'taken over' or 'invaded' islands that were once used by terns for nesting. Twenty seven percent of contributors (3 out of 11) also said that the presence of predators can prevent terns from laying eggs.

This little island used to be totally full of so many [tern] eggs you would hardly have to step a few feet to start gathering eggs [. . .] Nowadays, the terns are still there, but this summer they were not laying eggs. We have found only seagull eggs on that island [. . .] I think they were reluctant to lay eggs there because there was a colony of seagulls living there already [. . .] Terns can be overrun by seagulls. (Interviewee #1)

**Decline in Arctic Tern abundance around Kuujjuaraapik.** When describing Arctic Tern abundance and changes occurring over time, contributors referred to four key indicators of tern abundance: (1) number of terns flying around when going boating near nesting islands (i.e., seeing less terns flying around means there are fewer terns); (2) number of nests or nest density observed on nesting islands visited (i.e., observing fewer nests or lower nest density on nesting colonies means there are fewer terns); (3) effort required for egg picking or catch per unit effort (i.e., having to visit more islands to find enough eggs to practice selective egg picking means that there are fewer terns); and (4) level of noise made by terns when visiting colonies (i.e., the louder the noise, the more terns there are). Interviewees commonly used words such as 'empty', 'a few', 'some', 'not much', 'many', 'half full', 'full', 'everywhere' to express tern abundance in specific areas. When prompted, some contributors felt comfortable using orders of magnitude such as 'hundreds' and 'thousands' to describe how many terns they had observed on specific islands.

During individual and group interviews, all contributors reported a decline in Arctic Tern abundance on specific nesting islands around Kuujjuaraapik (Fig 4). One interviewee strikingly described:

The islands here [Ullugumitarviup Qikirtaarunga] used to be [. . .] totally full of so many terns that you could hardly see the sky or the sun sometimes. (Interviewee #1)

Sixty-four percent of contributors (7 out of 11) had observed an overall regional decline in tern abundance. Only one interviewee mentioned an increase in Arctic Tern numbers on specific nesting islands (i.e., Ivigainnaq and another island situated east of Ullugumitarviup Qikirtaarunga). Two contributors mentioned that Arctic Tern abundance was stable on specific islands (i.e., small islands situated near Cape Jones). Contributors referred to various timescales when describing regional or island-specific declines that they had observed. For example, interviewees used the following timescales to describe declines observed on specific nesting islands: decline for the past 20 to 25 years; decline for the past 20 years; decline for the past 10 to 20 years; decline for the past 10 years; decline for the past three years.

During the subsequent validation workshop (see methods section), a consensus emerged regarding the timing of the overall regional decline that had been observed by contributors: Arctic Terns were abundant between 1970s and 1990s, and a regional decline appeared in the early 2000s, with the most acute decline starting around 10 to 15 years ago. Some contributors discussed the difficulty of establishing the spatial scope of this regional decline given the limited geographic range of their observations (see Fig 2). Although contributors saw a decline specific to their region or specific nesting islands, they explained that terns could "have moved to an island that we didn't know" (Interviewee #4). While all contributors were questioned about whether they had observed disease or mortality events among Arctic Terns, only one reported seeing a few sick or dead young terns on one nesting island.

During individual and group interviews, contributors discussed six reasons that could explain why Arctic Terns had declined around their community. In total, 64% of contributors

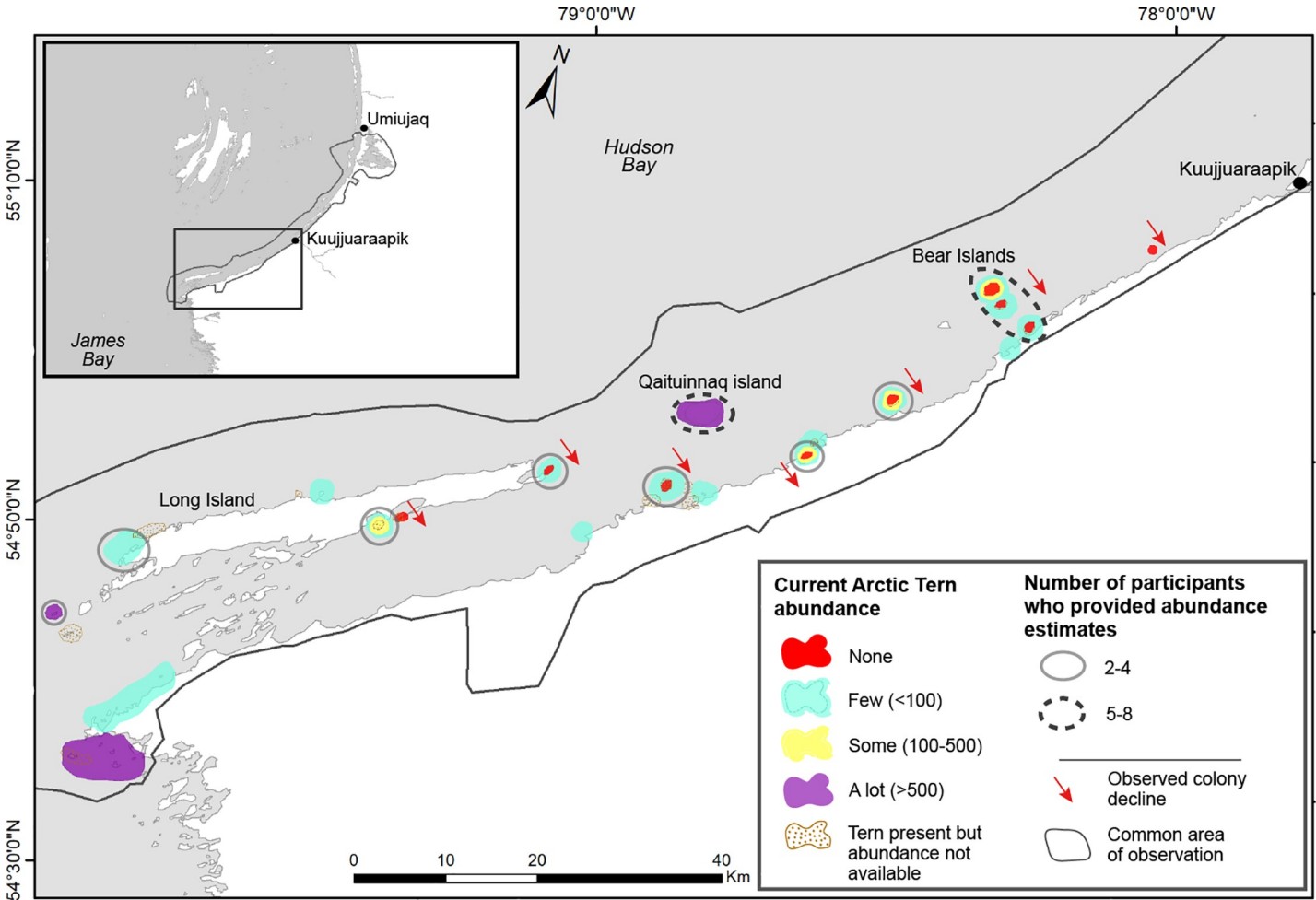

**Fig 4. Current Arctic Tern abundance and observed declines at specific nesting colonies according to interview contributors from Kuujjuaraapik (n = 11).** Coloured areas indicate current tern abundance identified by one or more contributors at specific nesting colonies. In case of colonies for which more than one abundance descriptor was used by contributors, multiple colours are used (e.g., Bear Islands shown in yellow [some terns], blue [few terns] and red [none]). Number of contributors who provided abundance estimates are indicated by the gray outlined circle (two to four contributors), the black dotted outlined circle (five to eight contributors) or no circle (one contributor). Red arrows indicate colonies where one or more contributors had observed a decline in recent years or decades.

(7 out of 11) reported local harvest as one of the possible causes for a regional decline. Some added that, over the past 10 to 15 years, the use of more powerful and faster motorboats could have contributed to an increase in tern egg harvest in the community.

> Overharvesting seems to be one of the main causes of the decline in the population. People would indiscriminately harvest, especially nearby. With faster motors, you can be there within 20 minutes. (Interviewee #1)

Disturbance and nest predation were also raised as a possible cause by 45% of contributors (5 out of 11).

> Up until recently this island [Ullugumitarviup Qikirtaarunga] was only a tern colony, but it's been slowly becoming [a seagull colony]. At one point, it was like half and half, but now it's only the seagulls that are laying eggs there [. . .] When terns are constantly being invaded, I think that's when they move to look for a better place to lay their eggs. (Interviewee #1)

It might be the polar bear population also where the polar bears are going to the islands [. . .] The Arctic Tern will not lay their eggs where there's a fox or a bear there. The polar bear population has something to do with it also. The Arctic Tern is taking all kinds of [hits] from all sides there, you know. (Interviewee #7)

Isostatic rebound, the rise of land masses after the retreat of large ice sheets produced during the last glacial period [45], was the third most reported possible cause of decline (36% of contributors; 4 out of 11). Interviewees explained that terns are no longer nesting on islands that have become connected to the mainland, as this connection brings terrestrial predators such as foxes.

No more [terns], not in the last 20 to 25 years because the land is rising. It used to be an island but now it's connected with the mainland. The Arctic Tern would lay eggs on the island but not mainland. (Interviewee #6)

In total, 27% of contributors (3 out of 11) reported the change in weather and climate as a potential cause of Arctic Tern decline. They explained that colder springs and summers observed in recent years on tern nesting grounds might contribute to the decline, as terns will not lay eggs if the weather is too cold. One interviewee reported that hurricanes can also lead to tern mortality along their migratory route although he had not observed this directly. Another mentioned that the weather had become more variable and less predictable over the last few decades.

[The decline] has something to do with our changing climate also [. . .] I noticed that one year there [were] so many [terns] and the following there were almost none. (Interviewees #6 and #7)

According to 27% of contributors (3 out of 11), tern abundance follows cycles, "and right now we might be in the low part of the Arctic Tern population cycle". The same percentage of interviewees reported a change in Arctic Tern prey as a possible cause of their decline. Capelin, which is an important food source for terns, has been reportedly less abundant in recent years. Two contributors suggested that this was likely due to warmer and less salty waters in Hudson Bay.

There's another reason why the tern population is so low it might be that what they're eating is not there anymore [. . .] It might be due to the temperature of the salt waters here [. . .] Our water used to be very salty and [it's] not anymore [. . .] It's impacting the terns' food [. . .] [Terns] lay their eggs in salty water where there's supposed to be capelin. (Interviewee #6 and #7)

Lastly, 55% of interviewees (6 out of 11) highlighted that a combination of factors is likely causing the observed decline. For example, according to one contributor, level of local harvest alone cannot explain everything given the sharpness of decline occurring in locations that are less frequently visited by people.

## Arctic Tern stewardship and management

**Past and ongoing stewardship practices.** Contributors discussed past and ongoing Arctic Tern stewardship practices. Some (27%; 3 out of 11) explained that they only took the eggs they needed and that they avoided wasting eggs. One mentioned the importance of showing respect to Arctic Terns, a value that has been passed down through generations.

I guess it was something that was passed down to us. "Don't hurt the little terns because they're going to provide you some eggs" kind of mentality. So we always respected that [. . .] We were told to respect them. (Interviewee #1)

In addition, 64% of contributors (7 out of 11) reported practicing selective Arctic Tern egg collection. Those who practiced selective egg picking explained that they would leave one or two eggs in the nest while collecting eggs to preserve the site for future egg picking, and not pick eggs in nests having three eggs or more due to the potential presence of embryos.

We used to only take what we needed because there was no point in taking more than what you needed if you're going to keep travelling. (Interviewee #1)

When we go, it's usually one or two [per nest], so I'll take them. But if there's more we leave one [. . .] We were always told to leave one behind. (Interviewee #9)

**Current level of concern.**   Interviewees discussed their overall level of concern about the observed decline of Arctic Tern abundance in the Kuujjuaraapik region. Those who had no concerns, corresponding to 55% of contributors (6 out of 11), explained that terns could adapt to changing environmental conditions or move elsewhere, and that tern eggs were not a main source of food in spite of being a local delicacy. Some also noted that the observed decline was probably part of the Arctic Tern population cycle, and, consequently, that Arctic Tern numbers would start to increase eventually.

I don't have any concerns with that [decline] [. . .] They're going to survive [. . .] They can adapt or move. (Interviewee #7)

Those who were concerned about the observed decline, corresponding to 45% of contributors (5 out of 11), were mainly worried about its impact on egg harvesting for present and future generations. Some expressed feelings of sadness when not seeing terns or not hearing their song.

[The decline] brings me a lot of concerns because I like to eat those eggs [. . .] And when there [are no eggs], you have to go further [to find some]. And this is what we had to do a couple of times, go further. (Interviewee #10)

I want [terns] back because they're part of this world and I have children and grandchildren and I would like them to live what I've lived, ate what I've eaten. (Interviewee #8)

One contributor further explained that harvesters in Kuujjuaraapik had discussed their concerns about the observed Arctic Tern decline, but there had been to date no concerted effort to restrict tern egg harvest by the community.

**Recommendations to support Arctic Tern stewardship and management.**   During individual and group interviews as well as the validation workshop, contributors expressed their opinions and perspectives about what should be done to support Arctic Tern stewardship and management in their region and beyond. They discussed eight specific recommendations: (1) conduct more research; (2) let nature take its course; (3) conduct an awareness campaign; (4) implement an egg collection ban; (5) coordinate local egg harvest; (6) start 'tern farming'; (7) protect Arctic Terns across their migration routes; and (8) harvest foxes that prey on terns.

A total of 82% of contributors (9 out of 11) recommended conducting additional research including expanded colony surveys, research on prey species and contaminants, migration tracking, and more interviews with Inuit experts from other Nunavik communities.

> If we can see a natural decline on a graph, maybe it will wake up some people to say this is a serious concern. It's not just our imagination [. . .] Survey, get actual numbers [. . .] [The decline is] actually happening right before our eyes. We just don't know how fast it's happening. (Interviewee #1)

> Around mid-July just do a survey [of nesting sites] [. . .] I think that would be very expensive though [. . .] and the weather is not always cooperating. (Interviewee #5)

> Little tracking [devices] would be nice [. . .] I know that they fly from far away to come up here. (Interviewee #9)

Importantly, among contributors who suggested conducting more research, some (18%; 2 out of 11) emphasized that additional research should not affect Inuit cultural practices such as tern egg picking, and stressed the importance of ensuring that no harm be done to terns during research.

> [A survey] would be good as long as it does not affect our way of life. (Interviewee #4)

The second most frequent recommendation (36% of contributors; 4 out of 11) was not to intervene and let nature take its course; in other words, not trying to change or influence current tern abundance or current egg harvesting practices. Twenty seven percent of contributors (3 out of 11) also recommended raising awareness among Nunavik Inuit about the observed Arctic Tern decline, and proposing protective measures through a communication campaign.

> An awareness campaign that the terns are in decline and refrain from picking for a few years [. . .] Have coordinated egg collecting maybe [. . .] because I'm pretty sure people are just taking what they need, but they don't know if people have been going [on the island where they pick from] before. (Interviewee #1)

Implementing a temporary ban on tern egg harvesting on nesting islands located nearby Kuujjuaraapik until terns recover was suggested by only 18% of contributors (2 out of 11).

> Jeez, I think something needs to be done [. . .] I think the opportunity is right there to make sure that [the LNUK of Kuujjuaraapik] do their duty and advise people that there's some concern of a decline of Arctic Tern: "Please refrain from picking eggs here" [. . .] I think some sort of a temporary ban should be placed on the nearby islands. (Interviewee #1)

On the contrary, three other contributors (27%; 3 out of 11) felt strongly that the local egg harvest should not be restricted. They explained that tern egg picking is an important cultural tradition which they want to maintain, and Kuujjuaraapik residents only take the eggs they need.

> We can't follow [restrictions on local egg harvesting]. Those are our eggs. How would you feel if I told you to stop eating chicken eggs? [Laughs] These are our once a summer delicacy! (Interviewee #8)

If I were told not to go [egg picking], I would definitely go anyway [. . .] I've gone every year for as long as I can remember. (Interviewee #9)

Coordinating the local egg harvest was proposed by one contributor to ensure that terns are not overharvested (i.e., that not too many people collect eggs from the same islands). Another interviewee suggested that the community starts 'tern farming' to keep predators away from tern nesting sites. Relatedly, one contributor recommended harvesting foxes that forage on Arctic Tern eggs to reduce pressure from predators. Finally, another contributor advised implementing protective measures across Arctic Terns migratory range, and not just in Kuuj-juaraapik or Nunavik regions.

If we were to do something about that population decline, you'll have to include all people [. . .] We can't do anything about it [alone]. (Interviewee #7)

## Discussion

Our interviews with Inuit harvesters provided important insights into the importance of Arctic Terns to Inuit communities, their knowledge of the ecology and population trends for the species near Kuujjuaraapik, and the factors they consider as important drivers of tern abundance. Importantly, contributors had observed declines in tern numbers consistent with reports from many other regions within the species' range [5, 8, 12, 46, 47]. Interviewees also identified some possible factors that may be influencing tern numbers, which were also reported for others species, such as increased predation on ground-nesting birds by polar bears [48] and increasing numbers of large gulls [49]. Here we discuss key study findings related to Inuit knowledge and perspectives on Arctic Tern cultural importance, ecology, stewardship and management recommendations, and contrast those with the existing literature on similar topics.

### Importance of Arctic Terns to Inuit

**Egg picking.**   Arctic Tern egg collection is important to Inuit from Nunavik in regards to their culture and community well-being. Arctic Tern eggs are not a main source of food for harvesters from Kuujjuaraapik but they are considered a local delicacy. Similarly, for Green-landers, harvesting Arctic Tern eggs is considered to be one of the highlights of the summer although not an important source of protein [7]. In the circumpolar Arctic, seabird egg harvesting by Indigenous peoples is an important cultural tradition, although its practice has diminished with increased access to alternative store-bought foods [50]. Egg picking around Kuujjuaraapik is often a multigenerational activity that requires cooperation among family and friends and mobilizes country food sharing networks [7, 51]. However, some Kuujjuaraa-pik residents seemed concerned that Arctic Tern knowledge was not being passed on to younger generations as it once was.

**Arctic Terns as indicators.**   The arrival of Arctic Terns around Kuujjuaraapik indicates to local residents that temperatures will be increasing, as in Greenland [7]. For Kuujjuaraapik harvesters, Arctic Terns can also indicate the presence of other harvested wildlife species, such as seals or fish. Similarly, in Alaska, Yu'pik hunters reported that Arctic Terns swooping over water indicates the presence of belugas, as terns feed on prey items forced to the surface by whales [52]. Non-Indigenous fishermen also use seabirds, which can rapidly cover large areas and locate prey efficiently, as indicators of commercial fish, such as capelin and lesser san-dlance (*Ammodytes marinus*) [53]. Around Kuujjuaraapik, Arctic Terns flight-diving over

nesting areas can alert Inuit harvesters to the presence of terrestrial mammals, including polar bears. Indeed, terns are infamous for their aggressive anti-predator behaviour within the boundary of their nesting colony [4, 7, 54].

## Arctic Tern ecology

**Breeding and predation.**   The timing of arrival, breeding and departure of Arctic Terns noted by Inuit contributors was consistent with reports from elsewhere in the circumpolar Arctic [4, 7, 9, 10, 12, 55], with variation attributable to latitude and local environmental conditions (details in S5 File). Inuit harvesters also provided information consistent with previous scientific studies on Arctic Tern predation [7, 10] (see S5 File). Interestingly, Inuit contributors reported that common ravens have only recently begun to feed on tern eggs in the Kuujjuaraapik region. Over the past decade, biologists have also observed ravens predating Arctic Terns in the High Arctic [7, 10].

**Feeding.**   Contributors observed that Artic Terns feed primarily on capelin in the Kuujjuaraapik region, similar to what has been reported for western Greenland [7]. Kuujjuaraapik residents explained that Arctic Terns can also feed on sandlance, which was reported to be the main prey of Arctic Terns in northern Scotland [5]. According to Kuujjuaraapik harvesters, sandlance were new to Hudson Bay, and it has been suggested that they may replace Arctic cod (*Boreogadus saida*) in the diet of seabirds elsewhere in Hudson Bay [9]. Invertebrates such as *Gammarus* spp. and mussels also contributed to the diet of Artic Terns around Kuujjuaraapik, as reported in other Arctic regions [7]. Around Kuujjuaraapik, Inuit harvesters reported seeing Arctic Terns feed in association with seals. In Alaska, Arctic Terns were reported by Yup'ik hunters to feed in association with belugas [52], and in Antarctica, biologists observed terns feeding in association with Antarctic minke whale (*Balaenoptera bonaerensis*) [12].

**Arctic Tern abundance around Kuujjuaraapik.**   Kuujjuaraapik residents unanimously reported that Arctic Tern abundance had declined on specific nesting islands around their community. Terns were reportedly abundant between the 1970s and the 1990s. A regional decline was first observed within contributors' common area of observation (Fig 2) in the early 2000s, with the most acute decline starting around 10 to 15 years ago. Similarly, over the past two decades, Inuit harvesters and biologists have reported a decline in Arctic Tern abundance in other regions of the Canadian Arctic, including the Belcher Islands in southeastern Hudson Bay [8], and along the coast of western Hudson Bay around the communities of Chesterfield Inlet (Charlotte Sharkey, pers. comm., 2019), Whale Cove (M. Mallory, unpubl. data), and Arviat (Natalie Carter, pers. comm., 2017). Declines of Arctic Terns have also been reported in northwestern Hudson Bay around the community of Coral Harbour on Southampton Island (Natalie Carter, pers. comm., 2017), the Nunatsiavut region (Amie Black, pers. comm., 2018), and in the Queens Channel, situated in the High Canadian Arctic [46]. Such observations are consistent with the decline of Arctic Terns reported worldwide [12]. For example, the greatest decline in northwest Greenland occurred within the past two decades, with an overall reported population decline of 50% since the 1960s-1990s [47]. In Europe, Arctic Tern numbers have apparently declined by approximately 25% since the 1980s [5, 12].

Contributors from Kuujjuaraapik who took part in this study identified six factors that could have potentially contributed to the decline observed in their region: (1) local harvest through egg picking; (2) nest disturbance and predation; (3) abandonment of tern nesting areas (i.e., islands that have become connected to the mainland due to isostatic rebound); (4) climate change; (5) normal abundance cycles within the Arctic Tern population; and (6) decline in Arctic Tern main prey in the region, capelin. Importantly, study contributors readily acknowledged the spatial limitations of their observations and, as such, mentioned that they

could have been unaware of some causes of decline happening outside of the coastal areas where they conducted land-based activities when terns are present in the region (Fig 2).

Some interviewees believed that Arctic Tern egg collection had increased in their community within the past 10 to 15 years due to the use of faster motorboats that enabled people to access nesting islands quicker, and that this could have contributed to the observed Arctic Tern decline. In Nunavik, motorized boats have been in use since the end of the 1960s (over 50 years ago), allowing Inuit harvesters to reach their hunting, fishing and trapping grounds and return within a day [56, 57]. Between 2006 and 2016, the Nunavik population has also grown by almost 30% [58]. An increase in Arctic Tern egg harvest in Kuujjuaraapik may therefore result from both faster boats being used by harvesters and local population growth generating greater demand for Arctic Tern eggs [58]. In Greenland, the population quadrupled since the 1950s, as did the number of motorized boats, and it has been suggested that declines in Arctic Tern nesting populations there could be explained by an unsustainable egg harvest conducted by local residents [7].

Kuujjuaraapik contributors also reported that nest disturbance and predation (by non-human predators) could possibly explain declines in Arctic Tern abundance. However, none of the contributors reported an increased in predator abundance or the arrival of new predators, except for the Common Raven. Similarly, Greenlandic Inuit have stated that the number of predators, which have always been present in Greenland, has increased, causing terns to decline [7]. Interestingly, one Kuujjuaraapik interviewee suggested that, as terns decrease in numbers, their ability to protect themselves against avian predators and protect their breeding sites from other nesting birds–including gulls, eider and Common Raven–declines (i.e., positive feedback loop). This process has been observed in colonies of another colonial nesting Arctic seabird, the Thick-billed Murre (*Uria lomvia*) [59]. Testimonies from contributors suggested that although the main reason of Arctic Tern decline might not be nest disturbance and predation per se, these factors may now increase reproductive failure of an already declining Arctic Tern population.

According to Kuujjuaraapik harvesters, isostatic rebound might explain part of the Arctic Tern decline observed at nesting colonies located around their community. Several islands that have historically supported Arctic Tern colonies have become connected to the mainland due to the rise of land masses following the melting of glaciers created during the last glacial period. Indeed, one of the fastest rising areas due to isostatic rebound in Canada corresponds to the southern coast of Hudson Bay, where Kuujjuaraapik is located [60]; the land in this area is currently rising by 12 millimetres per year or 1.2 meters per hundred years [60]. This might explain why contributors had witnessed some islands becoming connected to the mainland over the course of their lifetime. Terrestrial land bridges now enable predators such as Arctic foxes to reach island and Arctic Terns respond through abandonment of those breeding sites.

Contributors suggested that colder spring and summer observed over the last few years might contribute to the decline of Arctic Tern in their region. By contrast, other Nunavik residents–including those living in Umiujaq, located 160 km northeast of Kuujjuaraapik–reported less precipitation, earlier springs, longer and hotter summers than 30 to 40 years previously [61]. It is possible that, from year to year, there is a great variability in spring and summer temperatures [61]. Kuujjuaraapik contributors added that hurricanes can lead to tern mortality during migration. Although changes in temperature and snow cover conditions can lead to fluctuations in Arctic Tern numbers from year to year [10, 46], we did not find any information in the scientific literature linking changes in global or regional atmospheric temperatures to an observed Arctic Tern decline.

Interviewees also said that the decline in capelin, also reported in other parts of the Arctic [62, 63], might have contributed to a decrease in Arctic Tern abundance in their region.

Elsewhere there is strong evidence of changes in capelin or other food stocks influencing seabird breeding and numbers [5, 63–65]. Further research on capelin stocks in southeastern Hudson Bay is required to explore whether changes in capelin abundance in this region (if they have occurred), are influencing Arctic Tern foraging and reproductive ecology.

Finally, Kuujjuaraapik harvesters concluded that a combination of factors is likely causing the observed regional decline in Arctic Terns. A few contributors explained that Arctic Terns from some declining nesting islands could "have just moved elsewhere" (i.e., that their distribution had shifted). They also added that perhaps the decrease in Arctic Tern numbers on some islands is just part of the natural population cycle of Arctic Terns.

## Arctic Tern stewardship and management

**Past and ongoing stewardship practices.** Several Kuujjuaraapik contributors reported that they collect Arctic Tern eggs selectively. In contrast, Greenlandic harvesters were reported to collect all eggs from a clutch when a nest is found [7]. Contributors highlighted the capacity of Arctic Terns in Hudson Bay to lay replacement eggs following egg loss due to human harvest or natural predation. This confirms the results of an experiment conducted in Greenland demonstrating that Arctic Terns are able to produce replacement clutches with minimum effect on the overall productivity, as long as eggs are removed early in season [7].

**Recommendations to support Arctic Tern stewardship and management.** Kuujjuaraapik contributors discussed eight recommendations to support Arctic Tern stewardship in their region, although their perspectives and opinions varied. Among contributors who suggested conducting more research, some stressed the importance of ensuring that no harm is done to terns during research, thereby supporting the idea of 'showing respect' to terns, a value that has been passed through generations among Inuit from Kuujjuaraapik. Interestingly, according to a previous survey conducted in the Canadian High Arctic, Arctic Terns did not modify their nesting behaviour unexpectedly, possibly habituating to scientists who visited the colony on a daily basis [10], but overall behaving consistent with parental investment theory [66].

Some Kuujjuaraapik contributors suggested a total ban on egg collection in their community while others proposed a better coordination of egg harvest to ensure nesting colonies were not overharvested. While some harvesters from Kuujjuaraapik had discussed their concerns about the decline of Arctic Terns, there has been to date no concerted effort to restrict or coordinate tern egg harvest in the community. Among contributors, opinions regarding restrictions on egg harvesting differed markedly; while some believed restrictions should be implemented to prevent further decline, others advised not to intervene and to 'let nature take its course'. Some also felt strongly that the local egg harvest should not be restricted given that egg picking is an important cultural tradition which should be maintained and that residents only take the eggs they need. In Greenland, a ban on Arctic Tern harvesting was introduced in Greenland in 2002, resulting in local protests [7]. By contrast, harvest management ensuring that egg collections occurred early in the laying period had little affect on the population, as lost clutches were replaced [7]. Finally, some interviewees proposed harvesting foxes foraging on Arctic Tern eggs. This strategy has been implemented in Iceland, where there is intensive culling of foxes, gulls, and ravens near Arctic Tern nesting sites [7].

Overall, study contributors unanimously spoke of the importance of Arctic Terns to Nunavik Inuit for culture, community well-being, and sustenance, thus highlighting the need to ensure that tern eggs are sustainably harvested and remain available for present and future generations. In addition, many contributors highlighted the need to support knowledge sharing about Arctic Tern harvesting, ecology and stewardship with younger generations.

## Conclusion

Weaving together diverse knowledge systems generates an enriched picture of ecological issues of concern and broadens the spectrum of solutions that can support environmental decision-making and stewardship at multiple scales [67–69]. Collaborative environmental monitoring involving Inuit communities and scientific researchers can thus enhance our understanding of seabird ecology and inform wildlife co-management [22, 28]. As illustrated in this study, Nunavik Inuit possess broad knowledge of Arctic Terns, their habitat use, and their ecological interactions. Inuit knowledge offered long-term place-specific ecological observations that were generally consistent with or complementary to the scientific literature on Arctic Tern ecology; it also provided observations on Arctic Terns at a generally finer spatial resolution and over longer and more continuous time series than scientific research typically can. Combining Inuit knowledge with scientific knowledge can therefore increase the breadth and depth of our understanding of Arctic Tern ecology, and support management practices that are inclusive of Inuit views and recommendations.

Our study provides another example that Inuit harvesters can make substantial contributions to environmental research and monitoring efforts, as well as inform wildlife management practices. In the future, tern colonies identified in this study could be further monitored through a community-based program employing both scientific methods (e.g., colony surveys, migration tracking) and Inuit knowledge (e.g., using abundance indicators commonly employed by Inuit harvesters). Additional interviews could be conducted in other interested Nunavik communities, to expand the geographic scope of Inuit knowledge documented as part of regional population monitoring of Arctic Terns in Hudson Bay. Management recommendations suggested by Kuujjuaraapik contributors could also be further discussed and explored among Nunavik Inuit and other wildlife co-management partners to support Arctic Tern stewardship and sustainable use.

## Supporting information

**S1 File. Positionality of authors.**
(DOCX)

**S2 File. Interview guide for semi-directed interviews with Inuit experts.**
(DOCX)

**S3 File. Invitation letter and consent form.**
(DOCX)

**S4 File. Final results and data storage.**
(DOCX)

**S5 File. Additional discussion on Arctic Tern ecology.**
(DOCX)

## Acknowledgments

The authors would like to thank all Kuujjuaraapik residents who have generously shared their extensive knowledge and expertise as part of this project (in alphabetical order): Anthony Ittoshat, Billy Weetaltuk, Debra Weetaltuk, Samson Tooktoo, Jimmy Paul Angatookalook, Alec Tuckatuk, Moses Weetaltuk, and five anonymous contributors. We are grateful to the following individuals and organizations who have made this project possible: Shilo Weetaltuk (research assistant and interpreter), Eva Aloupa-Pilurtuut (translation), the Local Nunavimmi

Umajulivijiit Katujaqatigininga of Kuujjuaraapik, the Regional Nunavimmi Umajulivijiit Katujaqatigininga, the Nunavik Marine Region Wildlife Board, the Arctic Eider Society, and Environment and Climate Change Canada. We thank Joel Heath and SIKU/Arctic Eider Society for advice on project design, data sharing and archiving. We acknowledge the technical support of Debby Talbot with graphic design for Fig 2. We are grateful to Douglas Causey and one anonymous reviewer for their comments on an earlier version of this manuscript.

## Author Contributions

**Conceptualization:** Dominique A. Henri, Salamiva Weetaltuk, Mark L. Mallory, H. Grant Gilchrist, Frankie Jean-Gagnon.

**Data curation:** Dominique A. Henri, Frankie Jean-Gagnon.

**Formal analysis:** Dominique A. Henri, Laura M. Martinez-Levasseur, Frankie Jean-Gagnon.

**Funding acquisition:** Dominique A. Henri, Frankie Jean-Gagnon.

**Investigation:** Salamiva Weetaltuk, Frankie Jean-Gagnon.

**Methodology:** Dominique A. Henri, Mark L. Mallory, H. Grant Gilchrist, Frankie Jean-Gagnon.

**Project administration:** Dominique A. Henri, Frankie Jean-Gagnon.

**Resources:** Dominique A. Henri, Salamiva Weetaltuk, Frankie Jean-Gagnon.

**Software:** Dominique A. Henri.

**Supervision:** Dominique A. Henri.

**Validation:** Dominique A. Henri, Salamiva Weetaltuk, Frankie Jean-Gagnon.

**Visualization:** Frankie Jean-Gagnon.

**Writing – original draft:** Laura M. Martinez-Levasseur.

**Writing – review & editing:** Dominique A. Henri, Mark L. Mallory, H. Grant Gilchrist, Frankie Jean-Gagnon.

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
