## [Decision Letter · Decision Letter 0]

6 Aug 2020

PONE-D-20-19778

Inuit knowledge of Arctic Terns (*Sterna paradisaea*) and perspectives on declining abundance in southeastern Hudson Bay, Canada

PLOS ONE

Dear Dr. Henri,

Thank you for submitting your manuscript to PLOS ONE. After careful consideration, we feel that it has merit but does not fully meet PLOS ONE’s publication criteria as it currently stands. Therefore, we invite you to submit a revised version of the manuscript that addresses the points raised during the review process.

 Required changes - please address the comments provided by the 2 reviewers. Especially important are those that deal with figures and data presentation. In addition, the authors should consider a clear way distinguish traditional knowledge from area-specific observations as highlighted in the review.Please submit your revised manuscript by Sep 20 2020 11:59PM. If you will need more time than this to complete your revisions, please reply to this message or contact the journal office at plosone@plos.org. *Please include the following items when submitting your revised manuscript:*

*A rebuttal letter that responds to each point raised by the academic editor and reviewer(s). You should upload this letter as a separate file labeled 'Response to Reviewers'.**A marked-up copy of your manuscript that highlights changes made to the original version. You should upload this as a separate file labeled 'Revised Manuscript with Track Changes'.**An unmarked version of your revised paper without tracked changes. You should upload this as a separate file labeled 'Manuscript'.*

**

We look forward to receiving your revised manuscript.

Kind regards,

Christopher M. Somers

Academic Editor

PLOS ONE

*Journal Requirements:*

3. We note that Figure1, 2, 5 in your submission contain map images which may be copyrighted. All PLOS content is published under the Creative Commons Attribution License (CC BY 4.0), which means that the manuscript, images, and Supporting Information files will be freely available online, and any third party is permitted to access, download, copy, distribute, and use these materials in any way, even commercially, with proper attribution. For these reasons, we cannot publish previously copyrighted maps or satellite images created using proprietary data, such as Google software (Google Maps, Street View, and Earth). For more information, see our copyright guidelines: http://journals.plos.org/plosone/s/licenses-and-copyright.

3.1.    You may seek permission from the original copyright holder of Figure1, 2, 5 to publish the content specifically under the CC BY 4.0 license.

3.2.    If you are unable to obtain permission from the original copyright holder to publish these figures under the CC BY 4.0 license or if the copyright holder’s requirements are incompatible with the CC BY 4.0 license, please either i) remove the figure or ii) supply a replacement figure that complies with the CC BY 4.0 license. Please check copyright information on all replacement figures and update the figure caption with source information. If applicable, please specify in the figure caption text when a figure is similar but not identical to the original image and is therefore for illustrative purposes only.

**

**Reviewers' comments**:

*Reviewer's Responses to Questions*

***Comments to the Author***

*1. Is the manuscript technically sound, and do the data support the conclusions?*

*The manuscript must describe a technically sound piece of scientific research with data that supports the conclusions. Experiments must have been conducted rigorously, with appropriate controls, replication, and sample sizes. The conclusions must be drawn appropriately based on the data presented. *

*Reviewer #1: Yes*

*Reviewer #2: Yes*

*2. Has the statistical analysis been performed appropriately and rigorously? *

*Reviewer #1: N/A*

*Reviewer #2: N/A*

*3. Have the authors made all data underlying the findings in their manuscript fully available?*

*The PLOS Data policy requires authors to make all data underlying the findings described in their manuscript fully available without restriction, with rare exception (please refer to the Data Availability Statement in the manuscript PDF file). The data should be provided as part of the manuscript or its supporting information, or deposited to a public repository. For example, in addition to summary statistics, the data points behind means, medians and variance measures should be available. If there are restrictions on publicly sharing data—e.g. participant privacy or use of data from a third party—those must be specified.*

*Reviewer #1: Yes*

*Reviewer #2: Yes*

*4. Is the manuscript presented in an intelligible fashion and written in standard English?*

*PLOS ONE does not copyedit accepted manuscripts, so the language in submitted articles must be clear, correct, and unambiguous. Any typographical or grammatical errors should be corrected at revision, so please note any specific errors here.*

*Reviewer #1: Yes*

*Reviewer #2: Yes*

*5. Review Comments to the Author*

*Please use the space provided to explain your answers to the questions above. You may also include additional comments for the author, including concerns about dual publication, research ethics, or publication ethics. (Please upload your review as an attachment if it exceeds 20,000 characters)*

*Reviewer #1: Henri et al.’s manuscript documents interviews with Inuk people discussing their life-experienced knowledge of Arctic terns. They provide a wealth of natural history and phenology-based information about the local terns. The general information provided by these interviews is consistent with the literature, suggesting that this source of place specific information is generally reliable, and may provide a finer grain for this geographic area than has been provided by specific studies in different geographic areas. This information may support generalizing some scientific findings with Inuk observations from specific unstudied areas. Additionally, this source of information provides helpful ecological and temporal data regarding arrival and timing of new to the area species such as sandlance.*

Some place-specific observations over time, like the increase in use of fast motorboats to access islands, may allow for more informed interpretations of colony declines, since study by scientists is often only over short time spans. A lifetime of general observations is better for some place-based understanding than snapshot 1 to 4 field season studies by scientists. The two sources of knowledge lead to more full understanding when combined, for instance, observations by contributors of islands becoming attached to land masses combined with scientific data on rates of isostatic rebound.

While this provides useful information for conservation biologists and scientists that are planning studies of breeding arctic terns, it did not seem like Traditional Ecological Knowledge for the most part, although Local Ecological Knowledge does apply. It has not been passed down from an earlier generation to these hunters/fishers. Therefore, I suggest removing “Traditional” from the list of keywords. Nonetheless, place-specific long-term data from local hunters/fishers provide important and different data combined with the more generalized and shorter term data collected by scientists. Additionally, such anecdotal observations from contributors also leads to hypotheses to be tested more systematically by scientists. I think the manuscript would be improved by a statement in the discussion or conclusion that reflects the powerful combination of long-term place-specific data plus short-term generalized scientific data.

Specific points (some including line numbers)

Pattern of decline seems to stop some distance from Kuujjuaraapik except for a colony on Qaituinnaq Island. Is this island protected from humans? Why is it thought that this island colony not in decline?

Many of the quotes seem like personal knowledge based only on the speakers personal experience rather than traditional local ecological knowledge (historical, passed on from knowledge keepers).

I wonder if a review from an accepted and respected knowledge keeper might be helpful. Perhaps that was part of the verification meeting.

280-281 implies that they sometimes lay two separate successful clutches, which seems unlikely to me, given incubation time (at least 21 days) plus time to fledge at least 30 days. I suppose there could be some overlap of broods, but that would be a heavy task to be caring for 2 broods at the same time. Perhaps this observation is a case of adoption of the older chick. Second clutches after a nest failure seems likely.

352-354 Not a very sustainable view of egging.

681-683. Please clarify this statement to consider their reaction to intruders. I would presume that the birds either intensified their attacks or habituated to intruding scientists over the course of this study.

*404 – 406 Was anything said about the relative egg laying by gulls versus terns. In southern Ontario, gulls arrive and start breeding well before terns (Caspian and common) hence taking up space that would otherwise have been used by terns.*

*Reviewer #2: Overall, your MS is quite well done in presenting the results of your interviews with Inuit local people on their observations and Inuit knowledge relating to the population dynamics of Arctic Terns. The MS section "Arctic Tern ecology" was well handled as a comparison between local observations and our recorded scientific knowledge. The expanded review of Tern biology contained in Supplement 5 provided a more substantial overview of what's known as it relates to this very localized breeding population.*

Figure 2 seems redundant to what is presented in Figure 5. You could expand the caption in Figure 5 relating to this area within the grey line with the caption in Figure 2.

Figure 3: I suggest that the month "Septembre" should be presented in English "September" rather than French, to be consistent with the other months. August shows no harvesting and land activity...is that accurate?

*Figure 5: There is no green dotted line in the figure that is identified in the caption box. I think you refer to the thin dark line? Also, the printed caption (line 382) refers to the grey line, which does appear in Figure 2 but not here. These should be clarified.*

*6. PLOS authors have the option to publish the peer review history of their article (what does this mean?). If published, this will include your full peer review and any attached files.*

**

*Reviewer #1: No*

*Reviewer #2: **Yes: **Douglas Causey*

**

*While revising your submission, please upload your figure files to the Preflight Analysis and Conversion Engine (PACE) digital diagnostic tool, https://pacev2.apexcovantage.com/. PACE helps ensure that figures meet PLOS requirements. To use PACE, you must first register as a user. Registration is free. Then, login and navigate to the UPLOAD tab, where you will find detailed instructions on how to use the tool. If you encounter any issues or have any questions when using PACE, please email PLOS at figures@plos.org. Please note that Supporting Information files do not need this step.*

---

## [Author Response · Author response to Decision Letter 0]

27 Oct 2020

Dear Reviewers/Editors:

We would like to thank you for your constructive comments on our manuscript, and for this opportunity to revise our paper. We have now addressed all points raised during the review process and are pleased to submit a revised manuscript.

We have copied below all comments received from the academic editor and reviewers. Our responses (numbered) are presented after each comment. As specifically requested, we have thoroughly addressed all comments related to figures and data presentation, as well as identified a clear way to distinguish traditional knowledge from area-specific observations throughout our manuscript.

Please note that we also made sure to address all additional journal requirements in our submission (i.e., related to style, copyrighted material, and ethics). We present our comments related to additional journal requirements after our responses to reviewers/editor.

Sincerely,

Corresponding author (on behalf of all co-authors)

********

REVIEWERS’ COMMENTS

Reviewer's Responses to Questions

Comments to the Author

1. Is the manuscript technically sound, and do the data support the conclusions?

Reviewer #1: Yes

Reviewer #2: Yes

2. Has the statistical analysis been performed appropriately and rigorously? 

Reviewer #1: N/A

Reviewer #2: N/A

3. Have the authors made all data underlying the findings in their manuscript fully available?

Reviewer #1: Yes

Reviewer #2: Yes

4. Is the manuscript presented in an intelligible fashion and written in standard English?

Reviewer #1: Yes

Reviewer #2: Yes

5. Review Comments to the Author

Reviewer #1: Henri et al.’s manuscript documents interviews with Inuk people discussing their life-experienced knowledge of Arctic terns. They provide a wealth of natural history and phenology-based information about the local terns. The general information provided by these interviews is consistent with the literature, suggesting that this source of place specific information is generally reliable, and may provide a finer grain for this geographic area than has been provided by specific studies in different geographic areas. This information may support generalizing some scientific findings with Inuk observations from specific unstudied areas. Additionally, this source of information provides helpful ecological and temporal data regarding arrival and timing of new to the area species such as sandlance.

Some place-specific observations over time, like the increase in use of fast motorboats to access islands, may allow for more informed interpretations of colony declines, since study by scientists is often only over short time spans. A lifetime of general observations is better for some place-based understanding than snapshot 1 to 4 field season studies by scientists. The two sources of knowledge lead to more full understanding when combined, for instance, observations by contributors of islands becoming attached to land masses combined with scientific data on rates of isostatic rebound.

While this provides useful information for conservation biologists and scientists that are planning studies of breeding arctic terns, it did not seem like Traditional Ecological Knowledge for the most part, although Local Ecological Knowledge does apply. It has not been passed down from an earlier generation to these hunters/fishers. Therefore, I suggest removing “Traditional” from the list of keywords. Nonetheless, place-specific long-term data from local hunters/fishers provide important and different data combined with the more generalized and shorter term data collected by scientists. Additionally, such anecdotal observations from contributors also leads to hypotheses to be tested more systematically by scientists. I think the manuscript would be improved by a statement in the discussion or conclusion that reflects the powerful combination of long-term place-specific data plus short-term generalized scientific data.

Response from co-authors #1a: We thank you very much for your thoughtful and constructive review of our manuscript. 

We agree with the reviewer that most ecological observations presented in our manuscript were first-hand/direct observations that were not passed down from an earlier generation. As suggested by the reviewer, we have therefore removed “Traditional” from our list of keywords. The list of keywords now reads: “Inuit; Local Ecological Knowledge Canadian Arctic; Nunavik; Arctic Tern; decline.” 

In addition, as advised by the academic editor, we have now distinguished in a clear and systematic way throughout our manuscript (a) area-specific direct ecological observations made by contributors (Local Ecological Knowledge) from (b) observations/perspectives passed down from earlier generations (Traditional Ecological Knowledge). Our results section now reads: “When presenting results, we separated factual observations made by interviewees from inferences, and direct observations made by contributors from those they reported from other hunters [28,44]. Similarly, we distinguished contemporary knowledge from knowledge passed down from earlier generations.” (lines 170-172) 

We have now highlighted all information identified by contributors as coming from earlier generations. For example: “One interviewee spoke about the importance of showing respect to Arctic Terns, a value that has been passed down through generations.” (lines 204-205)

We also very much welcome the reviewer’s suggestion to highlight more explicitly the strength of combining TEK/LEK and science in our conclusion. We have modified our conclusion accordingly:

“Weaving together diverse knowledge systems generates an enriched picture of ecological issues of concern and broadens the spectrum of solutions that can support environmental decision-making and stewardship at multiple scales [67,68,69]. Collaborative environmental monitoring involving Inuit communities and scientific researchers can thus enhance our understanding of seabird ecology and inform wildlife co-management [22,28]. As illustrated in this study, Nunavik Inuit possess broad knowledge of Arctic Terns, their habitat use, and their ecological interactions. Inuit knowledge offered long-term place-specific ecological observations that were generally consistent with or complementary to the scientific literature on Arctic Tern ecology; it also provided observations on Arctic Terns at a generally finer spatial resolution and over longer and more continuous time series than scientific research typically can. Combining Inuit knowledge with scientific knowledge can therefore increase the breadth and depth of our understanding of Arctic Tern ecology, and support management practices that are inclusive of Inuit views and recommendations. 

Our study provides another example that Inuit harvesters can make substantial contributions to environmental research and monitoring efforts, as well as inform wildlife management practices. In the future, tern colonies identified in this study could be further monitored through a community-based program employing both scientific methods (e.g., colony surveys, migration tracking) and Inuit knowledge (e.g., using abundance indicators commonly employed by Inuit harvesters). Additional interviews could also be conducted in other interested Nunavik communities, to expand the geographic scope of Inuit knowledge documented as part of this study to support further regional population monitoring of Arctic Terns in Hudson Bay. Management recommendations suggested by Kuujjuaraapik contributors could also be further discussed and explored among Nunavik Inuit and other wildlife co-management partners to support Arctic Tern stewardship and sustainable use.” (lines 702-723) 

Specific points (some including line numbers)

Pattern of decline seems to stop some distance from Kuujjuaraapik except for a colony on Qaituinnaq Island. Is this island protected from humans? Why is it thought that this island colony not in decline?

Response #1b: The Qaituinnaq Island is not protected from humans – some harvesters from Kuujjuaraapik were able to provide abundance estimates from recent trips they had taken to this island to pick tern eggs. Harvesters estimated current Arctic tern abundance on Qaituinnaq Island at over 500 nesting terns, and contributors interviewed did not offer any explanation as to why this island colony was not in decline while neighbouring colonies were. Our tern researchers have noted similar odd responses elsewhere; for example, the colony where research has been conducted in the High Canadian Arctic (Mallory et al. 2017) has stayed at a relatively constant size despite declines in surrounding colonies. A variety of factors undoubtedly influence local tern colony dynamics, but those were not a focus of this study.

Many of the quotes seem like personal knowledge based only on the speakers personal experience rather than traditional local ecological knowledge (historical, passed on from knowledge keepers). I wonder if a review from an accepted and respected knowledge keeper might be helpful. Perhaps that was part of the verification meeting.

Response #1c: Yes, the reviewer is right. Most quotes of the quotes presented in our manuscript reflect personal experience and ecological observations from individual local experts. All individuals interviewed were identified and recommended by members of the local Hunters, Fishers and Trappers Association (HFTA) as local experts on Arctic Terns. In Kuujjuaraapik, members of the local HFTA (or LNUK) are recognized as subject matter experts and respected knowledge keepers when it comes to wildlife in their region. HFTA members were the ones who identified respected knowledge keepers who should be interviewed in their community. As part of our validation workshop, we presented in person our preliminary results to most interviewees and some HFTA members using a report and PowerPoint presentation. Interviewees and HFTA members therefore had an opportunity to discuss any collected information that they felt was unclear, incomplete or inaccurate. We also presented final project results to the community of Kuujjuaraapik in the form of a final project report (in English and Inuttitut) and comments received to date from project partners and knowledge keepers were very favorable. Importantly, one of our co-authors, Salamiva Weetaltuk, is a respected knowledge keeper and Inuk leader from Kuujjuaraapik; she has served on many local and regional wildlife boards over the years (including as Chairperson of the Nunavik Marine Region Wildlife Board), which speaks to her authority and leadership on wildlife matters in Nunavik. Ms. Weetaltuk has reviewed and contributed to our manuscript. As such, we think that respected local knowledge keepers have already verified results presented in our manuscript. We also fully agree with the reviewer having our results and analysis reviewed by accepted and respected local knowledge keepers (as a form of local peer-review) is very important to ensure the robustness of this type of work.

280-281 implies that they sometimes lay two separate successful clutches, which seems unlikely to me, given incubation time (at least 21 days) plus time to fledge at least 30 days. I suppose there could be some overlap of broods, but that would be a heavy task to be caring for 2 broods at the same time. Perhaps this observation is a case of adoption of the older chick. Second clutches after a nest failure seems likely.

Response #1d: We have reanalyzed the entire interview transcript for this interviewee, and we are confident that the person quoted here did not imply that terns could lay separate clutches. During our interview, this contributor explained that terns typically lay two to three eggs in one summer and that, in early fall, he could tell which chicks were younger and which were slightly older by looking at the size of their feathers. We agree however that the expressions “first round”, “second round”, and “third round” used in the quote can be confusing – please keep in mind that the interviewee expressed himself in Inuttitut and that English translation could have somewhat distorted original meaning here. We have therefore shortened the quote to avoid any confusion: “They normally have two or three eggs during the summer […] Just by looking at them [in early fall] and their small down feathers that they have, they tell you exactly how young or how old they are. (Interviewee #1)” (lines 278-280) We think this quote is valuable because it provides an example of how precise and detailed Inuit observations can be.

352-354 Not a very sustainable view of egging.

Response #1e: We recognize that this sentence requires some nuancing. As written, it suggests that harvesters would indiscriminately harvest tern eggs. On the contrary, most contributors reported practicing selective egg picking and highlighted how they would never pick all the eggs from a single colony. Those who practiced selective egg picking explained that they would leave one or two eggs in the nest while collecting eggs to preserve the site for future egg picking, and not pick eggs in nests having three eggs or more due to the potential presence of embryos.

We have therefore changed wording of this sentence to: “…(3) effort required for egg picking or catch per unit effort (i.e., having to visit more islands to find enough eggs to practice selective egg picking means that there are fewer terns)…” (lines 351-353). We do think this indicator of tern abundance is a very important one from a harvester perspective as it can indicate changes in species distribution and/abundance in a specific area.

681-683. Please clarify this statement to consider their reaction to intruders. I would presume that the birds either intensified their attacks or habituated to intruding scientists over the course of this study.

Response #1f: Good point. We have clarified the statement as follow: “Interestingly, according to a previous survey conducted in the Canadian High Arctic, Arctic Terns did not modify their nesting behaviour unexpectedly, possibly habituating to scientists who visited the colony on a daily basis [10], but overall behaving consistent with parental investment theory [66].” (lines 679-682) 

The study we are referring to here was conducted by Mallory et al. (2016, 2017) on Nasaruvaalik Island in Nunavut – note that Dr. Mallory is a co-author on our manuscript and has informed our response here. Scientists visited the Nasaruvaalik tern colonies twice daily over the course of five field seasons. They walked from the camp to the colony several times daily (except in periods of rain or heavy fog), arriving at blinds that were >50 m from the main nesting concentration of birds. Generally, their arrival did not cause an obvious change in behavior of the birds during nesting and terns habituated to the presence of biologists over time. Specific to nest defense, the birds left their nests earlier as incubation proceeded, exactly as one would predict for a ground-nesting bird in an exposed habitat.

404 – 406 Was anything said about the relative egg laying by gulls versus terns. In southern Ontario, gulls arrive and start breeding well before terns (Caspian and common) hence taking up space that would otherwise have been used by terns.

Response #g: Interviewees who commented on tern-gull interactions generally explained that some islands that used to have tern colonies were “invaded” or “overrun” by gulls preying on tern eggs and, as a result, that terns would not lay eggs on those islands anymore. One contributor also explained that gulls breed before terns around Kuujjuaraapik – gulls they lay their eggs in late May or June. So it does look like the same phenomenon maybe at play here, although this interviewee did not explicitly say that gulls were taking space that would otherwise have been used by terns. He suggested, however, that as terns decrease in numbers, their ability to protect themselves against avian predators and protect their breeding sites from gulls and other nesting birds declines (positive feedback loop). We made sure to include this important observation at lines 341-345. Timing of gull vs. tern breeding and more specific interactions between terns and gulls would definitely be an interesting topic to explore further with Inuit harvesters in the future.

Reviewer #2: Overall, your MS is quite well done in presenting the results of your interviews with Inuit local people on their observations and Inuit knowledge relating to the population dynamics of Arctic Terns. The MS section "Arctic Tern ecology" was well handled as a comparison between local observations and our recorded scientific knowledge. The expanded review of Tern biology contained in Supplement 5 provided a more substantial overview of what's known as it relates to this very localized breeding population.

Response from co-authors #2a: We are glad you appreciated the way we presented our results and compared Inuit knowledge with recorded scientific information. We thank you for your review and constructive comments.

Figure 2 seems redundant to what is presented in Figure 5. You could expand the caption in Figure 5 relating to this area within the grey line with the caption in Figure 2.

Response #2b: We agree with the reviewer that Figure 2 and 5 were somewhat redundant. have therefore completely removed our original Figure 2 and merged its content into Figure 1. As suggested by the reviewer, we have now expanded the caption for Figure 1 to include more information about the ‘common area of observation’. Caption for Figure 1 now reads: 

“Fig 1. Study area and common area of Arctic Tern observation among interview contributors from Kuujjuaraapik (n=11). The area located within the thick grey line indicates contributors’ common area of observation, i.e. area where interviewees conducted land-based activities (boating, camping, fishing, hunting, and egg picking) in summer and early fall, when terns were present in the region, and for which they had direct observational knowledge and experience.” (lines 110-114)

Figure 3: I suggest that the month "Septembre" should be presented in English "September" rather than French, to be consistent with the other months. August shows no harvesting and land activity... is that accurate?

Response #2c: We have corrected the “Septembre” typo, and added “Berry picking” for the month of August. Thank you for flagging this omission. We have also slightly changed caption wording for Figure 3 (now relabeled Figure 2) to highlight how harvesting and land-based activities presented here are only those relevant to tern observation by Inuit harvesters: “Fig 2. Arctic Tern breeding and migration phenology calendar. Information on Arctic Tern phenology provided by Kuujjuaraapik contributors (n=11) is presented above the timeline. The two rows under the timeline highlight associated environmental conditions and land-based activities relevant to tern observation by Inuit harvesters around Kuujjuaraapik. Timing of Arctic Tern life cycle events, environmental conditions and land-based activities can show interannual variations.” (lines 261-265) Lastly, we have completely redesigned the Figure to improve its overall look, and added drawings under each of the harvesting and land-based activities mentioned in the Figure.

Figure 5: There is no green dotted line in the figure that is identified in the caption box. I think you refer to the thin dark line? Also, the printed caption (line 382) refers to the grey line, which does appear in Figure 2 but not here. These should be clarified.

Response #2c: Thank you for pointing out to this error. The green dotted line in the caption box was replaced by a thick grey line representing contributors’ common area of observation. The information presented on the printed caption is now consistent with the information presented on the figure caption box. Please note that Figures 5 and 2 have been merged and relabeled Figure 2 (see also response #2b). 

********

JOURNAL REQUIREMENTS

Response from co-authors: Our manuscript meets PLOS ONE’s style requirements.

Response from co-authors: Data from this study should be available upon request as there are ethical restrictions on sharing data publicly. We have explained those ethical restrictions in detail in our revised cover letter and provided contact information for the institution to which data requests may be sent. 

Data from this study consist in audio recordings of interviews, interview metadata, interview transcripts, and information recorded on topographic maps. Data contain potentially sensitive information regarding Inuit land-use and harvesting activities, and use restrictions apply to this data through a data sharing agreement signed by all project partners.

3. We note that Figure1, 2, 5 in your submission contain map images which may be copyrighted. All PLOS content is published under the Creative Commons Attribution License (CC BY 4.0), which means that the manuscript, images, and Supporting Information files will be freely available online, and any third party is permitted to access, download, copy, distribute, and use these materials in any way, even commercially, with proper attribution. For these reasons, we cannot publish previously copyrighted maps or satellite images created using proprietary data, such as Google software (Google Maps, Street View, and Earth). For more information, see our copyright guidelines: http://journals.plos.org/plosone/s/licenses-and-copyright.

3.1. You may seek permission from the original copyright holder of Figure1, 2, 5 to publish the content specifically under the CC BY 4.0 license.

Comment from co-authors: All figures presented in our revised manuscript do not require permission to be published under the CC BY 4.0 license as they contain no copyrighted information. All figures presented in our manuscript are original (designed by our team) and have not been published elsewhere. We have now updated all figure captions with relevant source information.

---

## [Editor Report · Decision Letter 1]

29 Oct 2020

Inuit knowledge of Arctic Terns (*Sterna paradisaea*) and perspectives on declining abundance in southeastern Hudson Bay, Canada

PONE-D-20-19778R1

*Dear Dr. Henri,*

*We’re pleased to inform you that your manuscript has been judged scientifically suitable for publication and will be formally accepted for publication once it meets all outstanding technical requirements.*

*Within one week, you’ll receive an e-mail detailing the required amendments. When these have been addressed, you’ll receive a formal acceptance letter and your manuscript will be scheduled for publication.*

*An invoice for payment will follow shortly after the formal acceptance. To ensure an efficient process, please log into Editorial Manager at http://www.editorialmanager.com/pone/, click the 'Update My Information' link at the top of the page, and double check that your user information is up-to-date. If you have any billing related questions, please contact our Author Billing department directly at authorbilling@plos.org.*

*If your institution or institutions have a press office, please notify them about your upcoming paper to help maximize its impact. If they’ll be preparing press materials, please inform our press team as soon as possible -- no later than 48 hours after receiving the formal acceptance. Your manuscript will remain under strict press embargo until 2 pm Eastern Time on the date of publication. For more information, please contact onepress@plos.org.*

*Kind regards,*

*Christopher M. Somers*

Academic Editor

*PLOS ONE*

* *

*Additional Editor Comments (optional):*

* *
---

## [Editor Report · Acceptance letter]

3 Nov 2020

PONE-D-20-19778R1 

Inuit knowledge of Arctic Terns (*Sterna paradisaea*) and perspectives on declining abundance in southeastern Hudson Bay, Canada 

Dear Dr. Henri:

I'm pleased to inform you that your manuscript has been deemed suitable for publication in PLOS ONE. Congratulations! Your manuscript is now with our production department. 

Kind regards, 

on behalf of

Dr. Christopher M. Somers 

Academic Editor

PLOS ONE